# The Wound Healing Peptide, AES16-2M, Ameliorates Atopic Dermatitis In Vivo

**DOI:** 10.3390/molecules26041168

**Published:** 2021-02-22

**Authors:** Myun Soo Kim, Jisun Song, Sunyoung Park, Tae Sung Kim, Hyun Jeong Park, Daeho Cho

**Affiliations:** 1Korea University Kine Sciences Research Institute, Kine Sciences, 525, Seolleung-ro, Gangnam-gu, Seoul 06149, Korea; mskim@kinesciences.com (M.S.K.); sypark@kinesciences.com (S.P.); 2Division of Life Sciences, College of Life Sciences and Biotechnology, Korea University, 5-ga, Anam-dong, Seongbuk-gu, Seoul 02841, Korea; wltjs529@hanmail.net (J.S.); tskim@korea.ac.kr (T.S.K.); 3Institute of Convergence Science, Korea University, 5-ga, Anam-ro 145, Seongbuk-gu, Seoul 02841, Korea; dermacmc@naver.com

**Keywords:** short peptide, AES16-2M, atopic dermatitis

## Abstract

Peptide materials have recently been considered for use in various industrial fields. Because of their efficacy, safety, and low cost, therapeutic peptides are studied for various diseases, including atopic dermatitis (AD). AD is a common inflammatory skin disease impairing the patient’s quality of life. Various therapies, such as treatments with corticosteroids, calcineurin inhibitors, and antibody drugs, have been applied, but numerous side effects have been reported, including skin atrophy, burning, and infection. In the case of antibody drugs, immunogenicity against the drugs can be a problem. To overcome these side effects, small peptides are considered therapeutic agents. We previously identified the small wound healing peptide AES16-2M with a sequence of REGRT, and examined its effects on AD in this study. Interestingly, the administration of AES16-2M downregulated the AD disease score, ear thickness, serum IgE, and thymic stromal lymphopoietin (TSLP) in AD mice. The thickness of the epidermal layer was also improved by AES16-2M treatment. In addition, quantities of IL-4-, IL-13-, and IL-17-producing CD4 T cells from peripheral lymph nodes and spleens were reduced by injection of AES16-2M. Furthermore, the expression of TSLP was significantly reduced in AES16-2M-treated human keratinocytes. Therefore, these results suggest that AES16-2M can be a novel candidate for AD treatment.

## 1. Introduction

Peptide materials are currently interesting prospects as biomaterials that can be applied in various fields, including cosmetics, food, tissue repair, and therapeutics [1,2]. Chemically synthesized peptides have advantages for industrial uses, such as good safety, relatively low costs, and standard synthetic protocols [3,4]. Particularly in terms of therapeutic uses, numerous studies have identified functional peptides that can replace protein drugs with immunogenicity [4].

Atopic dermatitis (AD) is a chronic and common inflammatory skin disease caused by various genetic and environmental factors, and the worldwide prevalence of AD reaches up to 20% in the pediatric population [5,6,7,8]. AD patients suffer from itching, which can cause skin scratches, infections, and sleep disturbance [7,9]. Moreover, the social and emotional stress caused by the time and economic costs of treating the disease, and the damages to appearance, impair quality of life [7,9]. Because AD is an inflammatory disease mediated by cytokines such as IL-4, IL-13, IL-17, and thymic stromal lymphopoietin (TSLP) [10], topical corticosteroids and calcineurin inhibitors are applied as first-line therapy for AD patients in order to reduce inflammation [8]. However, their potential side effects, such as impaired skin barrier function, atrophy, burning, pruritus and infection, have been reported [8,11]. Although various antibody drugs, targeting T helper type 2 (Th2) cytokines and IgE, have been developed and used for AD treatment [12], protein drugs are relatively expensive, and are restricted in their long-term use as anti-drug antibody production reduces their efficacy [13,14]. To overcome these problems, functional peptides have recently been considered therapeutic agents because of their benefits of efficacy, safety, and low cost [3,4]. In particular, the potential immunogenicity of short peptides composed of less than 10 amino acids is considered very low, since the average size of the peptides loaded in MHC class II is over 10 amino acids in length [15].

We previously discovered a small peptide AES16-2M, with a sequence of REGRT, which shows dermal wound healing effects. AES16-2M was shown to induce ERK activation in keratinocytes and enhanced wound repair which is comparable to the respective effects of TGFβ and EGF [16]. Because both TGFβ and EGF attenuate AD in vivo and ERK activation in keratinocytes regulates skin inflammation [17,18,19], we hypothesized that the ERK-activating peptide, AES16-2M, could also ameliorate AD.

In this study, the therapeutic effects of AES16-2M were examined in the house dust mite allergen-induced AD model. The results show that the administration of AES16-2M significantly reduced the dermatitis score, ear thickness, and the level of blood IgE and TSLP in AD mice. The dermal tissue analysis also revealed that the AD-related thick epidermal layer was improved by AES16-2M treatment. In addition, we observed the production of IL-4, IL-13, and IL-17 in CD4 T cells from peripheral lymph nodes and spleens following the injection of AES16-2M. Furthermore, the expression of TSLP, a key factor for AD pathogenesis [20], was significantly decreased by AES16-2M in human keratinocytes. Therefore, these results suggest that AES16-2M could be a novel candidate for AD treatment.

## 2. Results

### 2.1. AES16-2M Ameliorates Dermatophagoides Farina Body-Induced AD in Mice

The properties of the AES16-2M peptide with 617.5 Da, pH 10.39 to 10.9 iso-electric point, 1.0 net charge at pH7.0, 1.7 average hydrophilicity, and over 10.0 mg/mL of water solubility, have been previously described [16] and are updated in this study. To investigate whether the wound healing peptide, AES16-2M, could improve AD, *Dermatophagoides farina* (Df) body-induced AD model was established, and 10 mg/kg of AES16-2M was used as the optimal dose determined by preliminary experiments. Surprisingly, AES16-2M administration attenuated AD symptoms and decreased the dermatitis scores significantly in AD mice, which results were comparable to those for the positive control group (dexamethasone) (Figure 1A,B). The histological analysis of dermal tissue sections also revealed that the thickness of the epidermal layer and the dermal infiltration of mononuclear cells following AD induction were improved as well (Figure 1C–E). In addition, the ear thickness of the AES16-2M-treated mice was thinner than that of vehicle-treated mice (Figure 1F). Since increased IgE and TSLP production is closely related to AD pathology [12], the total IgE levels from sera were determined. As expected, the serum IgE and TSLP levels were significantly lower in AES16-2M-administrated mice, and these levels were comparable to those in the dexamethasone group (Figure 1G,H). These results demonstrate that AES16-2M ameliorates AD in vivo.

### 2.2. AES16-2M Reduces the Generation of Th2 and Th17 Cells in the AD Model

It is well known that Th2 cells, which secrete IL-4 and IL-13, and IL-17-producing Th17 cells, are involved in AD development [10,21]. For these reasons, the changes in the populations of Th2 and Th17 cells were examined in the spleens and draining lymph nodes of the AD mice. The flow cytometric results determined that either IL-4-, IL13-, or IL-17-producing CD4 T cells were increased by AD induction, and that these populations were smaller in the spleens and lymph nodes from AES16-2M treated mice (Figure 2). These results suggest that AES16-2M efficiently reduces the generation of Th2 and Th17 cells, and attenuates AD pathogenesis.

### 2.3. AES16-2M Downregulates TSLP Expression in Normal Human Keratinocytes In Vitro

As AES16-2M showed its efficacy in the AD animal model, the potential therapeutic effects of the peptide on human cells were examined next. Because AES16-2M showed EGF-like effects on skin wound healing in our previous report [16], and because EGF administration reduces the expression of TSLP in skin [18] (an initial factor for AD pathogenesis) [20], AES16-2M was applied to activated normal human epidermal keratinocytes (NHEKs) to evaluate TSLP expression. The expression of CCL20, and Th17-recruiting chemokine, was also determined because the Th17 population was decreased by AES16-2M (Figure 2). Real-time PCR analysis showed that TSLP expression was induced by tumor necrosis factor-alpha (TNFα) and interleukin-4 (IL-4) in NHEK while the induction was inhibited in the presence of AES16-2M (Figure 3A). CCL20 expression was downregulated, but the changes were not significant (data not shown). The inhibition was also confirmed via semi-quantitative PCR and visualized bands (Figure 3B). Accordingly, the results indicate that AES16-2M has potential therapeutic effects on AD with the regulation of TSLP expression in human keratinocytes.

## 3. Discussion

Functional peptides are considered novel candidates for drug development because of their advantages, including their efficacy, safety, and relatively lower costs and numerous studies have discovered bioactive peptides [3,4]. However, extremely short peptides with sequences of less than 10 amino acids, which are free of the immunogenicity problem [15], are rare. In the current study, we demonstrated that the short peptide AES16-2M, consisting of only five amino acids, attenuates AD symptoms, including dermatitis score, ear and epidermal layer thickness, sera IgE, and TSLP level in AD mice. AES16-2M was also shown to reduce the generation of Th2 and Th17 cells, which produce IL-4, IL-13, and IL-17. Furthermore, the inflammatory cytokine-induced TSLP expression in NHEKs was inhibited by AES16-2M. Although the quantities of Th2 cytokines were reduced in the spleens of AES16-2M treated mice (Figure 2), in vitro IL-4 production in the total splenocytes was not changed by the peptide (data not shown). Therefore, AES16-2M is not thought to regulate Th2 polarization directly in the secondary immune organs. Since AES16-2M downregulated the serum TSLP level in vivo, it could be involved in the decreased Th2 responses.

In our previous study, the reported wound healing effects of AES16-2M were comparable to those of EGF and TGFβ [16]. Many studies suggest that the wound healing capacity of skin is closely related to AD. Caspase-8 deficiency affects both skin wound repair and AD development via the induced inflammatory responses [22,23]. IL-4, one of the pivotal factors for AD development, reduces the wound-repair ability in dermal tissue [24]. In addition, several materials with wound healing abilities, such as EGF, TGFβ and quercetin, also have therapeutic effects on AD [17,18,25,26,27]. Consistent with these findings, AES16-2M attenuated AD symptoms, as well as improving wound healing [16].

It is also significant that AES16-2M induces wound repair through enhanced ERK phosphorylation in keratinocytes [16], because the inhibition of ERK aggravates inflammation in the skin [19] and ERK activation is a part of the EGF and TGFβ signaling pathways [28,29]. However, it is known that the phosphorylation of ERK also induces TSLP expression [18]. Since AES16-2M was shown to downregulate TSLP expression in NHEKs, the mechanisms of this should be studied further.

There are several advantages to the transdermal delivery of drugs, such as its avoiding of gastrointestinal degradation, its ease of application, and its direct effects especially on dermal diseases [30]. In this respect, AES16-2M seems to have a benefit in terms of skin penetration because of its very small size, consisting of just five amino acids sequence. It is also possible to enhance the dermal permeability of peptides easily by conjugation of fatty acid chains [31]. Therefore, AES16-2M could be a novel candidate for the topical application of AD skincare as a bioactive short peptide.

## 4. Materials and Methods

### 4.1. Peptide, Mice, AD Model and Cells

AES16-2M was synthesized and analyzed for purity, water solubility, and molecular weight by Anygen (Gwang Ju, Korea). Peptide with purity over 95% was used for this study.

AD was induced in mice as previously described [32]. Briefly, 4% of sodium dodecyl sulfate was applied to the shaved skin of six-week-old female Nc/Nga mice (Central Lab. Animal Inc., Seoul, Korea). Three hours post-SDS treatment, AD cream containing *Dermatophagosides farinae* (Df) body ointment (Biostir, Kobe, Japan) was applied topically on the shaved skin and ears (100 mg/mouse). The AD cream was administrated total 6 times in total over 3 weeks, and AES16-2M (10 mg/kg) or dexamethasone (2.5 mg/kg) was injected the day following AD cream treatment (S.C. injection, 6 times in total over for 3 weeks). Ten mice per group were used for the animal study. The mice were maintained in a specific pathogen-free facility and the experiments were performed according to the guidelines of the Korea University Institutional Animal Care and Use Committee (KUIACUC-2020-0022).

Normal human epidermal keratinocytes (NHEK) were purchased from Promo Cell (Heidelberg, Germany) and were cultured in keratinocyte growth medium 2 (Promo cell).

### 4.2. Antibodies

Anti-CD3ε (145-2C11), anti-CD28 (37.51), alexa488-conjugated anti-IL-4 (11B11) and PerCP-Cy5.5-conjugated anti-CD4 (RM4-5) were purchased from BD Biosciences (San Diego, CA, USA). The PE-conjugated anti-IL13 (eBio13A) was from eBioscience (San Diego, CA, USA). The APC-conjugated anti-IL-17 (TC11-18H10.1) was from BioLegend (San Diego, CA, USA).

### 4.3. Evaluation of AD Score, Ear Thickness, IgE, and TSLP

The severity of AD was evaluated via a score of 0 (none), 1 (mild), 2 (moderate), and 3 (severe) for each symptom, including erythema/hemorrhage, scarring/dryness, edema, and excoriation/erosion [33]. Ear thickness was determined using an electronic caliper. For serum ELISA, blood samples were collected from mice of dermatitis score-matched 3 to 5 mice per group by cardiac puncture on day 21, and the total IgE and TSLP levels were examined using ELISA (eBioscience, San Diego, CA, USA).

### 4.4. Tissue Section Analysis

Skin tissues were isolated from mice and fixed in 10% formalin solution (Sigma-Aldrich, St Louis, MO, USA). The tissue samples were embedded in an optimum cutting temperature (OCT) compound (Sakura Finetek, Torrance, CA, USA), sectioned with a cryotome (Leica CM3050S, Leica, Berlin, Germany) at 5 mm thickness, and attached to slide glasses. Hematoxylin and eosin (H&E) staining was performed on the slides using the standard protocol [34].

### 4.5. PCR Analysis

HEK cells (2.5 × 10^5^ cells/well) were seeded in a 24-well plate and activated with TNFa (10 ng/mL, Peprotech, Cranbury, NJ, USA) or TNFa plus IL-4 (50 ng/mL, Peprotech, Cranbury, NJ, USA) in the presence of AES16-2M (0 to 1000 ng/mL) for 24 h. After cultivation, the RNA was isolated from the HEK cells and TSLP expression was evaluated by quantitative PCR (qPCR, LightCycler^®^96, Roche, Indianapolis, IN, USA). Semi-qPCR was also performed and bands were visualized by Amersham Imager 600 (Amersham Biosciences, Uppsala, Sweden). The primers used for TSLP were 5′-GAGTGGGACCAAAAGTACCG-3′ (sense) and 5′-CCAGATAGCTAAGG-3′ (antisense).

### 4.6. Flow Cytometry

The spleens and peripheral lymph nodes from the dermatitis score-matched 5 mice per group were homogenized, and cells from 5 mice in the same group were mixed. Splenocytes or lymph node cells (2 × 10^6^ cells/well) were cultivated separately in a 24-well plate in the presence of a plate-bound CD3ε antibody (1 μg/mL) and a soluble CD28 antibody (1mg/mL) for 20 h. Then, PMA (50 ng/mL, Sigma-Aldrich), ionomycin (1 mg/mL, Sigma-Aldrich) and monensin (1 mg/mL, Sigma-Aldrich) were added and incubated for 4 h. After the incubation, the cell surface molecules were stained with antibodies at a 1:500 dilution in FACS buffer for 15 min at RT and fixed with Cytofix/Cytoperm solution for 20 min at 4 °C (BD Biosciences, San Diego, CA, USA). Intracellular staining was performed in Perm/wash buffer for 1 h at 4 °C (BD Biosciences). FACSCalibur with CellQuest software was used for the flow cytometric analysis with gating of the live cells (BD Biosciences).

### 4.7. Statistical Analyses

A non-paired and two-tailed Student’s *t*-test was performed to compare the experimental groups and control groups. *p*-values < 0.05 were considered to be statistically significant.

## Figures and Tables

**Figure 1 molecules-26-01168-f001:**
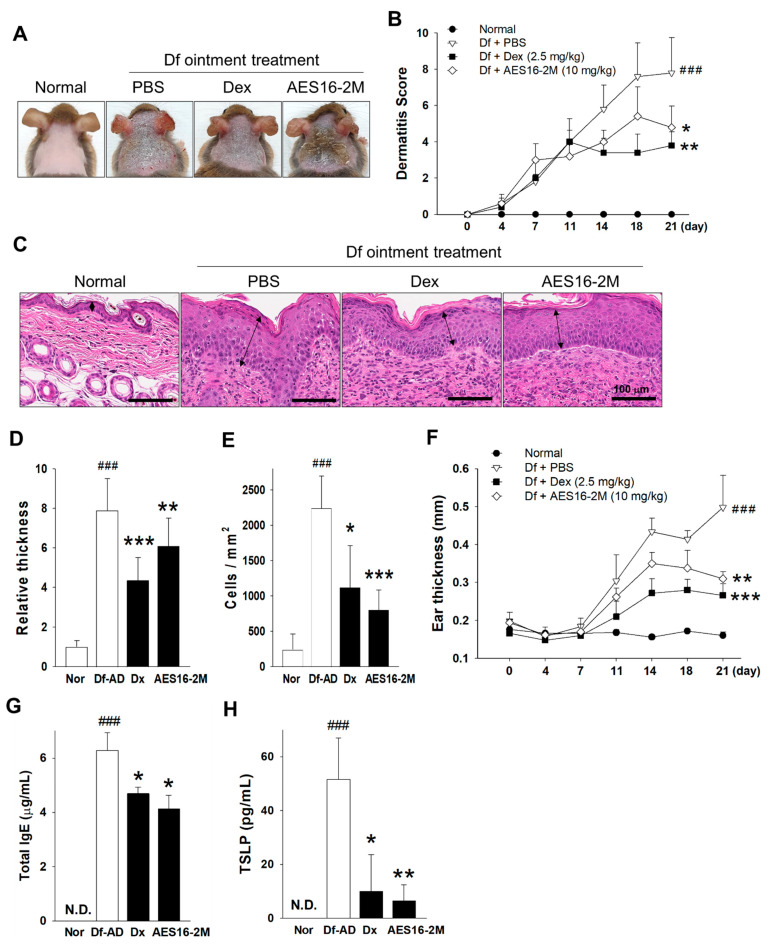
AES16-2M ameliorates *Dermatophagoides farina* (Df)-induced atopic dermatitis (AD). Df-induced AD was established as described in the Materials and Methods section. (**A**) The skin status was photographed on day 21. (**B**) The dermatitis score was compared between the AD mice and AES16-2M (10 mg/kg, S.C. injection) treated mice and summarized as mean ± SD (*n* = 10). Dexamethasone (Dex, 2.5 mg/kg) was used for the positive control group. (**C**) H&E stain of dermal tissues from day 21 was visualized by microscopy. Arrows reveal the epidermis. (**D**) The relative epidermal thickness was evaluated via data derived from (**C**) (three mice per group in quintuplicate). The thickness was measured using ImageJ and presented as mean relative values to the normal control. (**E**) Mononuclear cells that had infiltrated into the dermis were counted. (**F**) Ear thickness was measured using a digital caliper and was expressed as mean ± SD (*n* = 10). The total IgE (**G**) and TSLP (**H**) levels were evaluated from the sera of mice on day 21 (*n* = 5 for IgE, *n* = 3 for TSLP). The animal experiments were performed with a total of 10 mice per group. In the case of ELISA, sera from the dermatitis score-matched mice were used. A non-paired and two-tailed Student’s *t*-test was performed to compare the experimental groups and control groups. ###, *p* < 0.001 (vs normal); * *p* < 0.05 (vs PBS); ** *p* < 0.01 (vs. PBS); *** *p* < 0.001 (vs. PBS).

**Figure 2 molecules-26-01168-f002:**
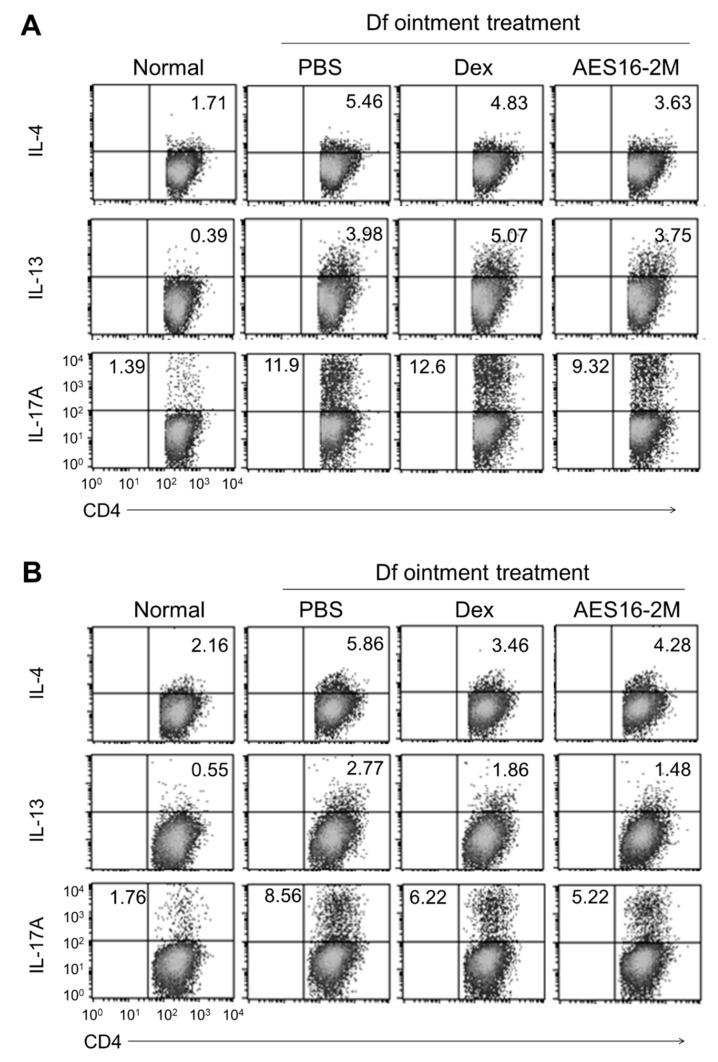
AES16-2M reduces the generation of either IL-4-, IL-13-, or IL-17-producing CD4 T cells in vivo. (**A**) Draining lymph nodes and (**B**) spleens were isolated from mice of dermatitis score-matched five mice per group, and the cells were cultured in the presence of TCR stimuli for 20 h (1 μg/mL of CD3εand CD28 antibodies). PMA (50 ng/mL), ionomycin (1 μg/mL) and monensin (1 μg/mL) were added into the cells and incubated for 4 h. The IL-4-, IL-13-, or IL-17-producing CD4 T cells were evaluated by flow cytometry (cells gated on CD4^+^). Dex, dexamethasone.

**Figure 3 molecules-26-01168-f003:**
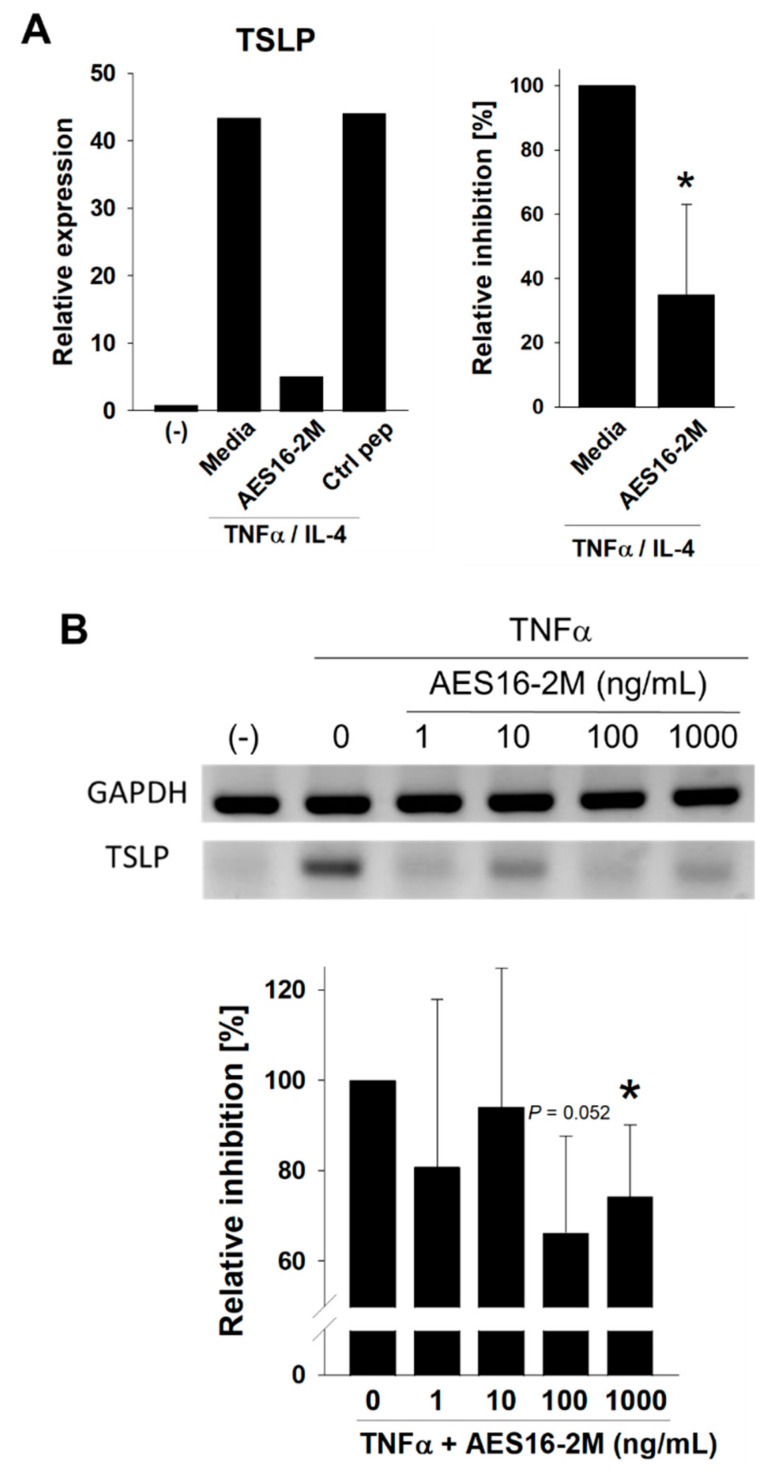
AES16-2M inhibits thymic stromal lymphopoietin (TSLP) expression in human keratinocytes in vitro. (**A**) Normal human epidermal keratinocytes (NHEKs) were activated with tumor necrosis factor-alpha (TNFα, 10 ng/mL) and interleukin-4 (IL-4, 50 ng/mL) in the presence or absence of AES16-2M (100 ng/mL). RNA was isolated and the TSLP expression levels were compared using quantitative PCR (left panel). The relative decrease assessed via three independent experiments is presented as a percentage of the TNFα/IL-4 control (right panel, *n* = 3). (**B**) TSLP expression was also confirmed using semi-quantitative PCR, and the visualized bands. The density of the band was evaluated using ImageJ, and relative decreases assessed via three independent experiments are presented as percentages vs. the TNFαcontrol (*n* = 3). For the relative decreases, the value from each set of the cytokine-treated control was presented as 100%, and the relative decreases were evaluated against the cytokine-treated controls. A non-paired and two-tailed Student’s *t*-test was performed to compare the experimental groups and control groups. * *p* < 0.05 (vs. not treated control).

## Data Availability

Data sharing not applicable.

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
