# Peer review of "The Wound Healing Peptide, AES16-2M, Ameliorates Atopic Dermatitis In Vivo"

_molecules, 2021, doi:10.3390/molecules26041168_

Round 1

Reviewer 1 Report

The authors shown that wound healing peptide AES16-2M improve skin inflammation and production of IgE in mice induced by Dermatophagosides farinae (Df) topical treatment. The improvement in skin inflammation was observed by improvement of clinical score, decreased ear swelling and lessen epidermal hyperplasia. In addition, the authors shown decreased percentages of CD4+ cells producing IL-13 and IL-17A in draining lymph node and spleen cells. In addition, the authirs shown that the peptide AES16-2M decreases TSLP mRNA levels in normal human epidermal keratinocytes stimulated by TNFa and IL-4.

Major comments.

The authors only shown only representative images of histology, flow cytometry plots and semi-quantitative PCR (n=1), the authors must show the results of all the analyzed mice/samples in graph to have an idea the mean responses in the groups and the dispersion of these responses.

The authors evaluate the significance by t test, however they did not specify which groups were compared in these analysis (PBS vs treatment?, PBS vs Normal? Treatment vs normal? AES16-2M vs Dex?). the statistical analysis of variance will be more adequate to have comparison in all the groups in the same test.

There are claims by the authors exaggerating the obtained results, such as “AD induction was improved dramatically”, “These results demonstrate that AES16-2M ameliorates AD excellently in vivo “. However, Dex treated mice showed more pronounced improvement of the disease.

Specific points.

The authors shown a representative image of H&E stained skin section, they need to quantify the epidermal thickness in all the groups and plot it in a graph to appreciate the  extent of the reduction in AES16-2M and compared it with the control treatment (Dex).

Tha authors claim that AES16-2M decreased the number of IL-4 producing CD4+ cells in the spleen and draining lymph node, however it is almost not detectable IL-4 production by CD4 cells in draining lymph node and spleen. The authors must improve the quality of the stimulation or staining to show more convincing data.

In the flow plots Dex treated mice show an increased production of IL-13 and IL-17A in draining lymph node cells, How the authors explain this increase?

The author may consider to evaluate the secretion of the cytokines measured after Df restimulation in vitro by ELISA 

The authors showed that AES16-2M decreased TSLP expression in normal human epidermal keratinocytes stimulated by TNFa and IL-4. The authors only analyzed one sample as not error bars are present in the bar graph. The authors suggest that the improvement of skin inflammation in mice is due to the decrease in TSLP expression in keratinocytes, however, the authors did not evaluate TSLP expression in the Df treated skin subjected to vehicle (PBS) and AES16-2M treatment. In addition, the authors injected AES16-2M subcutaneously, It will be interesting to evaluate whether AES16-2M topical treatment will induce decrease in TSLP expression and improvement of Df-induced skin inflammation.

Reviewer 2 Report

The authors describe the use of a small peptide AES16-2M, that is used for wound healing, to also be a potential therapetic for atopic dermatitis. They use AES16-2M in an murine model of AD and in some in vitro studies in human. AES16-2M appears to work be activating EGFR pathways that inhibit TSLP1 secretion, though this is not directly shown.

I have, however, a few questions:

1) It is not clear whether AE16-2M causes reduced TSLP1 by the EGFR pathway as the authors point out that ERK can also induce TSLP expression. It would be of interest to determine whether AES16-2M blocks phosphorylation of ERK in response to TNF/IL4 treatments performed in Figure 3B. The authors could simply blot for phospho-ERK with the samples. This could be complemented with an ERK inhibitor as a control.

EGF was shown to attenuate ERK phosphorylation and TSLP expression in an in vivo model of DNCB-induced AD (Ref 18).

This would increase the impact of the message of the paper.

2) There are no error bars in Figure 3A. The expt should be performed at least 3 times and the means graphed with the Standard error of the mean (SEM) and statistical analysis performed.

3) A TSLP ELISA should be performed to complement human studies in Figure 3.

4) While a reduction of TSLP would lead to reduced Th2 cell infiltrate, why do the authors see reduced Th17? The authors should perform qPCR / ELISA for CCL20 in samples from Figure 3. A reduction of CCL20 expresssion might explain decreased IL-17-producing cells.

5) Figure 2 would be helped with some statistical analysis. It is not shown how often expts are repeated and this should be corrected in the figure legends.

6) Similarly, Fig 1 is also missing description of the statistical analysis. Importantly, in Fig1E, what are the Dx and AES16-2M groups compared to? They should be compared to Df-AD but it’s not clear.

7) the reduced inflammation is not so easy to see in figure 1A. Can the authors provide better pictures?

8) epidermis and dermis should be labelled in 1C. with AES16-2M there looks to be less immune cell infiltrate than a reduction of epidermal thickening compared to DEX. The reduction could be more in the dermis than the epidermis.

9) in line 85. Change “excellently” to “significantly”, as it is comparable to the positive control. In line 155 change “noting” to “note”

10) Label all samples in 3A and 3B that have been treated with TNF/IL-4 with (+) or TNF/IL-4 as it looks like other samples weren’t treated from current labelling.

Reviewer 3 Report

This manuscript describes the in vivo efficacy of a small peptide AES16-2M in atopic dermatitis (AD) mice model, and the result indicated that AES16-2M reduced dermatitis score, ear thickness, and the level of blood IgE. Further studies indicated that AES16-2M reduced the generation of Th2 and Th17 cells and inhibited the expression of TSLP. Overall, the finding in this manuscript is interesting, and would be helpful to the relevant research field. However, major revisions are needed to address the following major concerns before acceptance for publication in Molecules.

  1. The manuscript needs a native English speaker to polish the language and improve the writing quality. There are numerous grammar mistakes and poorly presentations. For example: (1) “Peptide materials currently attract attention as biomaterials which can be applied to various fields, including cosmetics, food, tissue repair, and therapeutics [1, 2].” This sentence is uncompleted. (2) “Especially for therapeutic uses, numerous studies have identified functional peptides which can replace protein drugs with immunogenicity” This sentence is not right presented. It should be “… without immunogenicity” or “… with minimal immunogenicity”. (3) “Because AD is an inflammatory disease mediated by cytokines such as IL-4, IL-13, IL-17, and thymic stromal lymphopoietin (TSLP) [10], topical corticosteroids and calcineurin inhibitors are treated as first-line therapy for AD patients to reduce inflammation” The sentence is too long to make a clear understanding…

        …

        Not fully listed.

        (4) In line 55, “the” should be “a”. (5) In line 56 “has” should be deleted.

  1. “in vivo” should be italic as “in vivo”.
  2. “ml” should be “mL”.
  3. The “P” value should be added in the figure 1 legend.
  4. In figure 3A, the error bar should be included.
  5. In line 179, “with REGRT sequence” should be deleted. Also, the 1 mg/kg dose of AES16-2M should be deleted. It was not mentioned in the text.

Round 2

Reviewer 1 Report

I thank you the authors taking in consideration my comments, however some of my queries were not answered accurately.

The authors included in the revised version of the manuscript graph showing mean and SD of results obtained for epidermal thickness (1D) and semi-quantitative PCR (fig 3B). However, not description of how the epidermal thickness or band density was measured is included in the revised version of the manuscript. In addition, not standard deviation is shown in the first bar graph of figure 3B. The authors must include how they normalized the results and show the dispersion in the control group.

Regarding the intracellular staining, analyzing individual mice treated with AES16 could give a better idea how efficient is this treatment suppressing IL-13 or IL-17, with the results showed we could conclude that the treatment suppress IL-13 and IL-17 production by CD4+ T cells, but we do not know if the suppression is observed in the entire group (5/5 mice) or there is variability between the group (4/5 mice or 3/5 mice). 

Regarding the IL-4 staining, I am still not convinced that treatment decrease the production of IL-4 in CD4+ T cells. The authors must show the results for the IL-4 secretion after DF re-stimulation in vitro to strength this argument.

Regarding  TSLP levels in skin of Df treated mice, the authors shown that serum TSLP levels are decreased after AES16-2M, however, other TSLP sources, in addition of keratinocytes, will be responsible for serum TSLP levels. 

Reviewer 2 Report

The authors have answered my comments.

A human TSLP ELISA in Figure 3 would be welcome but I understand the problems with obtaining reagents in today's current climate and the TSLP ELISA in Figure 1 in the mouse model adds significant support to the findings.

A couple of minor points.:

The statistical tests used should be stated in the figure legends.

In Figure 3 A-C can the authors include TNF/IL-4 instead of the (+) in the figure? It makes it easier for the reader.

Author Response

  1. (Comment): The statistical tests used should be stated in the figure legends.

(Response): As the reviewer commented, we added the statistical information in the figure legends (Figure 1 and 3).

  1. (Comment): In Figure 3 A-C can the authors include TNF/IL-4 instead of the (+) in the figure? It makes it easier for the reader.

(Response): As the reviewer commented, we changed the labels in the figure (Figure 3).

Reviewer 3 Report

All the concens have been well addressed and the changes have improved the manuscript significantly. Therefore, my advice is acceptance. 

Author Response

The reviewer had no comment.